# Comparative Study on the Adsorption Capacities of the Three Black Phosphorus-Based Materials for Methylene Blue in Water

**Juanhong Wang †, Zhaocheng Zhang †, Dongyang He, Hao Yang, Dexin Jin, Jiao Qu * and Yanan Zhang**

School of Environment, Northeast Normal University, NO. 2555 Jingyue Street, Changchun 130117, China; wangjh468@nenu.edu.cn (J.W.); zhangzc363@nenu.edu.cn (Z.Z.); hedy443@nenu.edu.cn (D.H.); yangh409@nenu.edu.cn (H.Y.); jindx446@nenu.edu.cn (D.J.); zhangyn912@nenu.edu.cn (Y.Z.)
* Correspondence: quj100@nenu.edu.cn; Tel.: +86-431-89165617; Fax: +86-431-89165610
† These authors contributed equally to this work.

**Abstract:** Dye effluent has attracted considerable attention from worldwide researchers due to its harm and toxicity in recent years; as a result, the treatment for dye has become one of the focuses in the environmental field. Adsorption has been widely applied in water treatment owing to its various advantages. However, the adsorption behaviors of the new materials, such as the 2D black phosphorus (BP), for pollution were urgently revealed and improved. In this work, BP, black phosphorene (BPR), and sulfonated BPR (BPRS) were prepared by the vapor phase deposition method, liquid-phase exfoliating method, and modification with sulfonation, respectively. The three BP-based materials were characterized and used as adsorbents for the removal of methylene blue (MB) in water. The results showed that the specific surface areas (SSAs) of BP, BPR, and BPRS were only 6.78, 6.92, and 7.72 $m^2 \cdot g^{-1}$, respectively. However, the maximum adsorption capacities of BP, BPR, and BPRS for MB could reach up to 84.03, 91.74, and 140.85 $mg \cdot g^{-1}$, which were higher than other reported materials with large SSAs such as graphene (GP), nanosheet/magnetite, and reduced graphene oxide (rGO). In the process of BP adsorbing MB, wrinkles were generated, and the wrinkles would further induce adsorption. BPR had fewer layers (3–5), more wrinkles, and stronger adsorption capacity (91.74 $mg \cdot g^{-1}$). The interactions between the BP-based materials and MB might cause the BP-based materials to deform, i.e., to form wrinkles, thereby creating new adsorption sites between layers, and then further inducing adsorption. Although the wrinkles had a certain promotion effect, the adsorption capacity was limited, so the sulfonic acid functional group was introduced to modify BPR to increase its adsorption sites and promote the adsorption effect. These findings could provide a new viewpoint and insight on the adsorption behavior and potential application of the BP-based materials.

**Keywords:** black phosphorus; phosphorene; sulfonation; adsorption; methylene blue

## 1. Introduction

In recent years, with the rapid development of new textiles, printing and dyeing technologies have become more and more complicated. Dyes in effluent lead to the deterioration of water quality [1]. Dye pollution in water is often accompanied with high chromaticity, high organic content, high toxicity, and refractory degradation, which could cause a series of risks to the ecological system and human health [2,3]. Therefore, the problem of dye contaminants in global water has received more and more attention [4,5]. As dyes have been widely used in industrial production and daily routine,

the development of disposal methods for dye in wastewater has become a research hotspot in the water environmental protection field [6,7].

The widely applied adsorption technology mainly utilizes the porous structures and the active sites in the adsorbents for the removal of pollution [8,9]. It has many advantages, such as high adsorption efficiency, simplicity, and selective enrichment of certain compounds [10]. Therefore, it is particularly important to search for new adsorbents, illuminate their adsorption mechanisms for pollutants, and study modification methods [11].

As the rising stars of post-graphene (GP) two-dimensional (2D) nanomaterial, black phosphorus (BP) and black phosphorene (BPR) have received more and more attention. Due to the direct bandgap and high carrier mobility of BP, it is supplementary for the lack of GP's bandgap and the low carrier mobility of transition metal sulfide materials. In addition, its remarkable light absorption efficiency, plane anisotropy, and photoelectric properties give BP huge application potentials in the communication and energy fields [12,13]. BP is a conductive crystal with metallic luster and composed of phosphorus atoms [14,15]. In accordance with GP and transition metal sulfides, BP also has a similarly lamellar structure [16]. The layers are connected by strong intralayer P–P bonding and weak interlayer van der Waals forces [17,18]. BP has three structures including orthorhombic, cubic, and trigonal [19–21]. Orthogonally two-dimensional BP has a wrinkled surface and belongs to the lattice structure [22,23].

BPR has been applied to electronics, photonics, and photothermal/photodynamic therapy in biomedical research [24,25]. Recently, BPR has been explored to be used as a drug delivery platform [26,27]. The adsorption capacity of BPR for methylene blue (MB) was studied using the theoretical calculation based on the density functional theory (DFT); in addition, the adsorption isotherm and adsorption kinetics were also carried out systematically [28]. For the first time, the maximum theoretical specific surface area (SSA) of BPR was calculated to be 2400 $m^2 \cdot g^{-1}$, and BPR had a high adsorption capacity for MB ($1232 \pm 283$ $mg \cdot g^{-1}$). The results meant that BPR could be expected to be used as an efficient adsorbent for MB in aqueous solutions. The adsorption capacities of BPR, modified by polyethylene glycol, for doxorubicin and anthocyanins could reach up to 1080 and 304 $mg \cdot g^{-1}$ [29]. To our best knowledge, although BP has great application potentials in the environmental field, there were very few studies on BP and BPR, even fewer studies on the modification of BPR. In this work, we prepared BPR by the physical method, and then modified BPR by the chemical method to prepare BPRS. The SSA of BP was increased, thereby increasing the adsorption sites of BP, and finally increasing the adsorption capacity of BP for MB. Although the SSAs of the three BP-based materials were smaller than other adsorbents, their adsorption capacities for MB were higher than those of other adsorbents. The adsorption laws were explored by adsorption kinetics, adsorption thermodynamics, adsorption isotherms, desorption, the influence factors of temperature, and pH. The adsorption mechanisms were clarified through characterization and compared with other materials, which could provide a reference for the future application of BP-based materials in the environmental field.

Therefore, BP and BPR were prepared by the vapor phase deposition method and liquid-phase exfoliating method in this work. Subsequently, BPR was modified by sulfonation to form sulfonated BPR (BPRS) to improve its stability and adsorption capacity. Afterwards, the prepared BP, BPR, and BPRS were characterized and used as adsorbents to the removal of MB in water. In addition, the adsorption capacities and mechanisms of BP, BPR, and BPRS for MB were also investigated and compared.

## 2. Materials and Methods

### 2.1. Preparation of BP-Based Materials

#### 2.1.1. Preparation of BP

BP was prepared by the vapor phase deposition method [30] as follows: Firstly, 80 mg Sn, 30 mg $I_2$, and 500 mg red phosphorus (RP) were sealed in a quartz tube and placed flatwise in a muffle furnace after being vacuumized. Secondly, the typical temperature program was set as follows:

The temperature of the furnace increased to 600 °C and maintained for 2 h, decreased to 490 °C and maintained for 2 h, and decreased to 120 °C slowly. Finally, the quartz tube was taken out and cooled down to room temperature naturally [31].

### 2.1.2. Preparation of BPR

BPR was prepared by the liquid-phase exfoliating method [32,33]. Under the ice bath conditions, 100 mg BP and 100 mL N-methylpyrrolidone (NMP) were added into an ultrasonic cell pulverizer and exfoliated for 12 h [34,35]. Afterward, the supernatant was extracted after being centrifuged at 12,000 rpm for 10 min, and the precipitates were dried at 80 °C for 6 h after being washed 3 times with absolute ethanol.

### 2.1.3. Preparation of BPRS

The prepared BPR was modified by sulfonation to form BPRS. BPR was added into the aryldiazonium salt solution, mixed with 46 mg sulfanilic acid, 18 mg sodium nitrite in 10 g water, and 0.5 g 1 N HCl solution in the ice bath, and stirred for 8 h in the ice bath [36,37]. Afterwards, the supernatant was extracted after being centrifuged at 12,000 rpm for 10 min. In addition, the collected precipitates were washed once with absolute ethanol and dried at 80 °C for 6 h.

### 2.2. Characterization

Scanning electron microscope (SEM) images were carried out with a JEOL JSM-840 operated at 3.0 kV; energy dispersive spectrum (EDS) elemental mapping was performed with an Oxford EDX system attached to SEM; SSA and porosity were determined by Brunauer-Emmett-Teller (BET) from $N_2$ adsorption–desorption method (Micrometrics ASAP 2020M PLUS HD88); pore size distribution and volume were obtained from the $N_2$ adsorption isotherms by Barrett–Joyner–Halenda (BJH) method; X-ray diffraction (XRD) patterns were performed on a Rigaku D-max C III (monochromatic Cu K$\alpha$ radiation); Raman spectra were carried out using a micro-Raman spectrometer (Nicolet Almega XR) with a 532 nm laser as an excitation source; atomic force microscopy (AFM) images were performed using a Nanoscope III in tapping mode with a NSC14/no Al probe (Dimension icon, Veeco); fourier transform infrared (FTIR) spectra (4000–400 cm$^{-1}$) were conducted using a Nicolet 6700-FTIR spectrometer (Thermo Nicolet, Madison) equipped with a resolution of 0.125 cm$^{-1}$.

### 2.3. Adsorption for MB

The adsorption capacities of BP, BPR, and BPRS for MB in water were investigated. Batch adsorption experiments were performed in triangle vials sealed with foil paper. Considering the ionic strength of natural water, 1110 mg·L$^{-1}$ CaCl$_2$ was added to the background solution. In addition, 200 mg·L$^{-1}$ NaN$_3$ was used as the bioinhibitor [36]. The water sample prepared with MB was blue and toxic. The COD concentration was 115.4 mg·L$^{-1}$, the chromaticity was 1216.5 times, and the pH was 7.23. In the experiments, two duplicates were carried out.

The adsorption kinetic experiments were conducted with an initial MB concentration of 20 mg·L$^{-1}$; the solid-to-liquid ratio of BP, BPR, and BPRS to MB was 2 mg per 10 mL, and the adsorption time ranged from 1 to 120 h. Isotherm experiments were carried out with the same solid-to-liquid ratio, and the initial concentrations of MB solution ranged from 1 to 25 mg·L$^{-1}$. Thermodynamic experiments were performed with the same solid-to-liquid ratio, and the adsorption temperatures ranged from 298.15 to 318.15 K. The effects of adsorbent dosage (from 1 to 5 mg) and pH (from 2.70 to 10.68, adjusted with 0.1 M NaOH and 0.1 M H$_2$SO$_4$) on adsorption for MB were investigated. After the adsorption experiment, the desorption experiments were carried out directly at 308.15 K. The supernatant was removed by a pipette carefully; 10 mL water was added. The second desorption was repeated. To reach equilibrium, a series of independent samples were agitating in dark at 150 rpm on a thermostatic water bath shaker. The absorbance of MB in the supernatant was determined by UV-visible spectrum after being centrifuged at 12,000 rpm for 10 min.

## 2.4. Data Analysis

The adsorption experiments were carried out using a batch experimental technique, and the equilibrium adsorption capacity of the adsorbent was calculated using the following equation:

$$q_e = \frac{V(C_0 - C_e)}{m}$$

(1)

where $q_e$ (mg·g$^{-1}$) was equilibrium capacity of adsorption, $C_0$ and $C_e$ (mg·L$^{-1}$) were the initial and equilibrium concentrations of MB, $V$ (L) was the solution volume of MB, and $m$ (g) was adsorbent dosage.

The quasi-first-order kinetic model was presented as follows [38,39]:

$$ln(q_1 - q_t) = lnq_1 - k_1$$

(2)

where $k_1$ (h$^{-1}$) was the rate constant of the quasi-first-order kinetic model, $q_1$ and $q_t$ (mg·g$^{-1}$) were the adsorption capacities at equilibrium and at t time, and the values of $k_1$ and $q_1$ were determined from the slope and intercept of linear fittings of $ln(q_1 - q_t)$ versus $t$ (h).

The quasi-second-order kinetic model was shown as follows [40]:

$$\frac{t}{q_t} = \frac{1}{k_2 q_2^2} + \frac{t}{q_2}$$

(3)

where $k_2$ (g·(mg·h)$^{-1}$) was the rate constant of the quasi-second-order kinetic model, $q_2$ and $q_t$ (mg·g$^{-1}$) were defined the same as the parameters in the quasi-first-order kinetic model, and the values of $q_2$ and $k_2$ were determined from the slope and intercept of linear fittings of $t/q_t$ versus $t$. When the fitting of adsorption kinetics data was performed using the quasi-second-order kinetic model, the instantaneous driving force of adsorption was underestimated, leading to a false overestimation of the rate constant. In order to solve this problem, a correction method for model fitting was proposed. In this method, the rate constant was found to depend on the solute concentration and adsorption time [41].

The Langmuir model was shown as follows:

$$Ce/q_e = Ce/q_m + 1/K_L q_m$$

(4)

where $K_L$ (L·g$^{-1}$) was the Langmuir constant, and $q_m$ (mg·g$^{-1}$) represented the maximum adsorption capacity of the adsorbent.

The Freundlich model was shown as follows:

$$lnq_e = lnK_f + lnC_e/n$$

(5)

where $K_f$ ((mg·g$^{-1}$)·(mg·L$^{-1}$)$^{-n}$) was the Freundlich affinity coefficient, and $n$ was the exponential coefficient.

The Gibbs energy change ($\Delta G^0$) indicated the degree of the spontaneity of an adsorption process, and a higher negative value reflected more energetically favorable adsorption. According to thermodynamic analysis, $\Delta G^0$ of adsorption was calculated as follows:

$$\Delta G^0 = -RTlnK_c$$

(6)

where $\Delta G^0$ (KJ·mol$^{-1}$) represented adsorption free energy change, $R$ (8.314 J·(mol·K)$^{-1}$) was gas constant, $T$ (K) represented thermodynamic temperature, and $K_c$ was the thermodynamic equilibrium constant without units [42], which was calculated by the following formula:

$$K_c = \frac{q_e}{C_e}$$

(7)

Van't Hoff equation:

$$lnK_c = \frac{\Delta S^0}{R} - \frac{\Delta H^0}{RT} \tag{8}$$

where $\Delta H^0$ (KJ·mol$^{-1}$) represented adsorption enthalpy change, and $\Delta S^0$ (KJ·mol$^{-1}$) represented adsorption entropy change.

## 3. Results and Discussions

### 3.1. Characterization

SEM images of the prepared BP, BPR, and BPRS are shown in Figure 1a–c. The morphologies of the three prepared BP-based materials were block structure. Compared with the prepared BP, the BPR and BPRS were smaller owing to the broken structure after being ultrasonic exfoliated, which might result in a bigger SSA of BPR. From the EDS elemental mapping (Figure 1d), it could be observed that the primary elements in the prepared BPRS contained S, O, and P. Other elements including Cl, N, and C might come from impurities of RP and the used conductive adhesive. The result meant BPRS was obtained after BPR was successfully sulfonated.

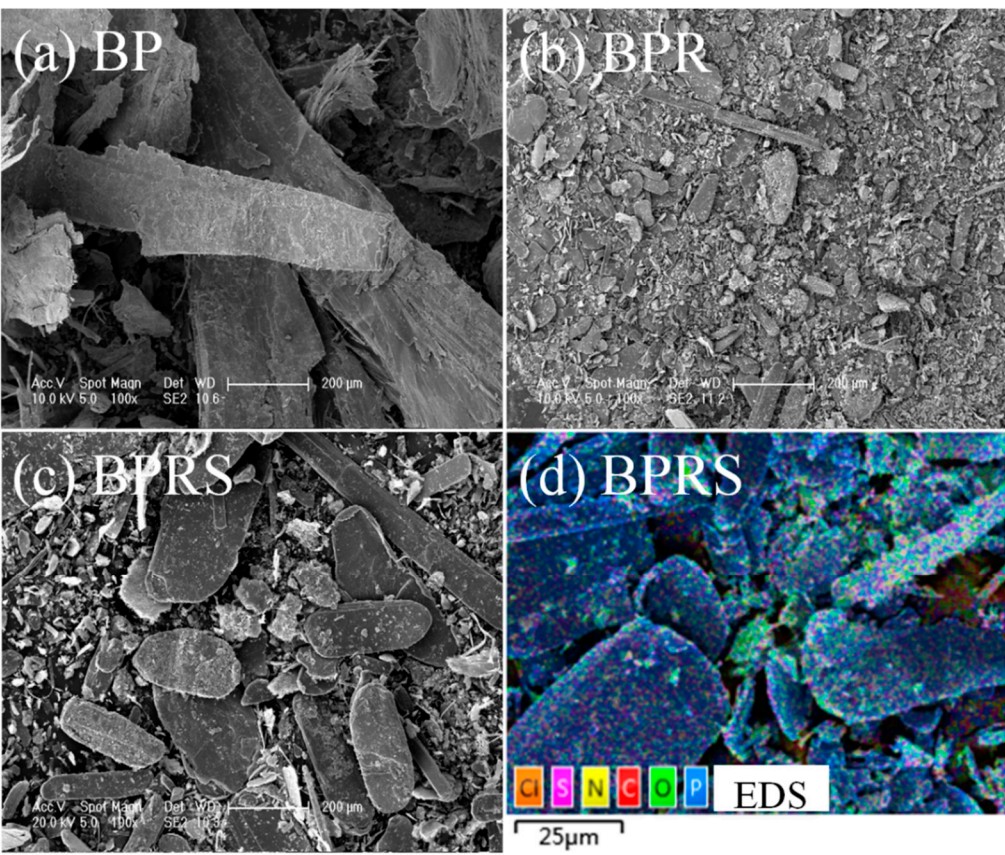

**Figure 1.** SEM images of (**a**) BP, (**b**) BPR, and (**c**) BPRS; (**d**) EDS elemental mapping of BPRS.

XRD patterns of the prepared BP, BPR, and BPRS are shown in Figure 2a. The diffraction peaks (020), (040), and (060) were corresponding to the orientation of BP (JCPDS no. 73–1358) [30,43]. The Raman spectra of the prepared BP, BPR, and BPRS are shown in Figure 2b. The peaks at 363.6, 440.7, and 467.7 cm$^{-1}$, caused by vibrations of P–P bonding, were attributed to $A_g^1$, $B_{2g}$, and $A_g^2$ modes of BP [44,45], respectively. As a result, it could be concluded that the three prepared materials belonged to BP-based material.

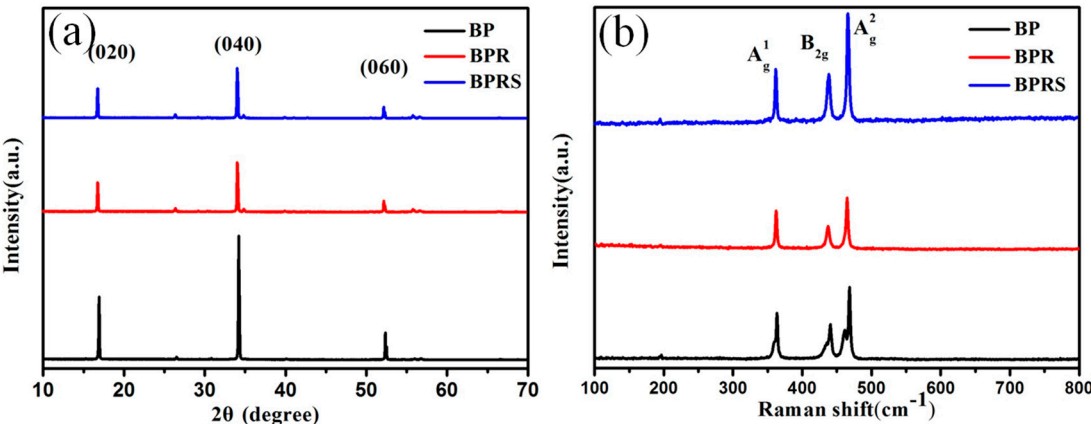

**Figure 2.** (**a**) XRD patterns and (**b**) Raman spectra of BP, BPR, and BPRS.

According to the previous reports, the thickness of monolayer BP was 0.53 nm [33,46]. As shown in Figure 3, the thicknesses of BP were 24.99 and 16.38 nm; the results meant that the prepared BP was bulk. The thicknesses of BPR were 1.32 and 2.15 nm, illustrating that its layer number was 3–5. Similarly, the thicknesses of BPRS were 2.48 and 2.81 nm, indicating that its layer number was 5–6. To sum up, the thicknesses of BPR and BPRS were thinner than BP, owing to the decreasing layer numbers of BP by exfoliating in the cell pulverizer.

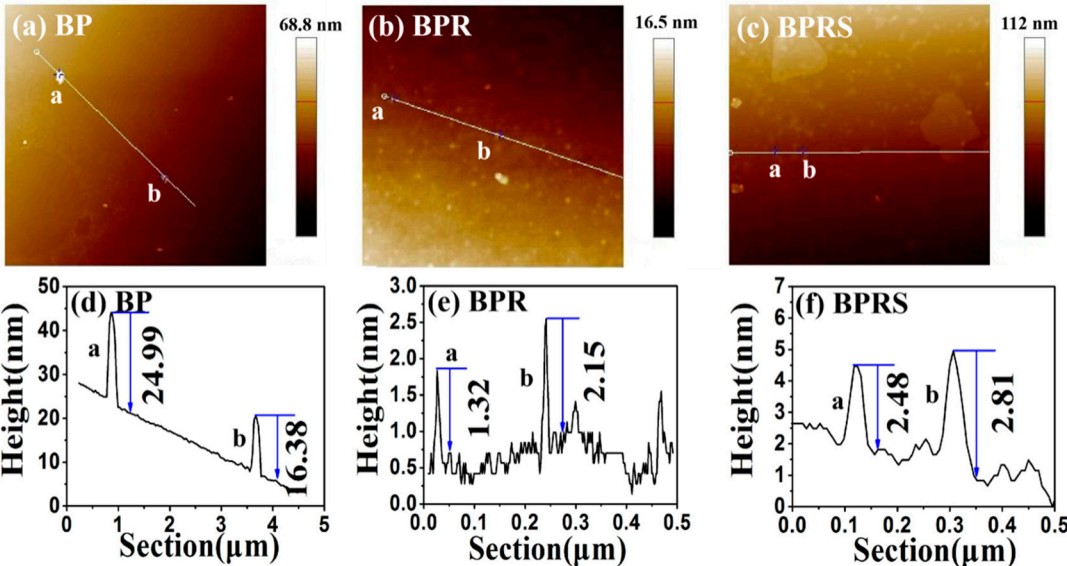

**Figure 3.** AFM images of (**a**) BP, (**b**) BPR, and (**c**) BPRS; statistical analysis of thickness for (**d**) BP, (**e**) BPR, and (**f**) BPRS.

As shown in Figure 4a and Table 1, all specific surface areas (SSAs) of the prepared BP, BPR, and BPRS were small. The SSA of BPR ($6.96 \text{ m}^2 \cdot \text{g}^{-1}$) was larger than that of BP ($6.78 \text{ m}^2 \cdot \text{g}^{-1}$), possibly owing to the exfoliation and hierarchical porosity [47,48]. The hierarchical pores were beneficial to the exposure of the BPR surface. However, the obtained SSA of BPR was much lower than the theoretical value ($2400 \text{ m}^2 \cdot \text{g}^{-1}$) [28], which was likely related to incomplete exfoliation and aggregation during the preparation process. The SSA of BPRS ($7.72 \text{ m}^2 \cdot \text{g}^{-1}$) was larger than that of BPR, because the lack of ionizable groups on the surface of BPR made it easy to stack via the hydrophobic effect, and then inhibited the active sites to be exposed. The charged $HSO_3^-$ groups on the surface of BPRS enhanced the dispersibility and prevented aggregation, resulting in less stacking. The pore size distribution curves were shown in Figure 4b. Compared to other reported adsorbents, the BJH pore volumes and BJH

average pore sizes of the prepared BP, BPR, and BPRS were also small and the results were also shown in Table 1. The BJH pore volume order of 0.0451 cm$^3 \cdot$g$^{-1}$ (BPRS) > 0.0319 cm$^3 \cdot$g$^{-1}$ (BPR) > 0.0303 cm$^3 \cdot$g$^{-1}$ (BP) indicated that BPRS had the largest pore volume. It meant BPRS had the most active adsorption sites, followed by BPR, and finally BP. The pore sizes of micropore, macropore, and mesopore were defined as less than 2 nm, greater than 50 nm, and between 2 and 50 nm, respectively [49]. As mesopore was conducive to the distribution of active sites of the adsorbent, it increased the contact between active sites and the adsorbate, which was beneficial for adsorption [50]. The BJH average pore size order was BP (13.99 nm) > BPRS (12.81 nm) > BPR (12.75 nm), and the pore sizes of the three BP-based materials ranged from 2 to 50 nm, indicating that the three BP-based materials belonged to excellent mesoporous adsorbents.

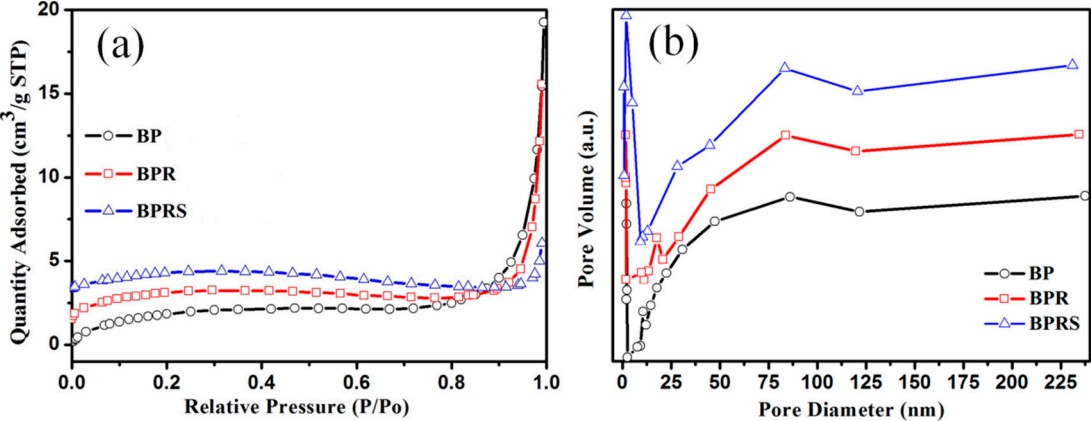

**Figure 4.** (**a**) N$_2$ adsorption isotherms and (**b**) pore size distributions.

**Table 1.** SSAs and pore structures of BP, BPR, and BPRS.

| Sample | SSA (m$^2 \cdot$g$^{-1}$) | Pore Volume (cm$^3 \cdot$g$^{-1}$) | Average Pore Size (nm) |
|--------|-------------|--------------|----------------------|
| BP   | 6.78 | 0.0303 | 13.99 |
| BPR  | 6.96 | 0.0319 | 12.75 |
| BPRS | 7.72 | 0.0451 | 12.81 |

In order to better understand the interaction between the three BP-based materials and MB, FTIR spectroscopy was introduced to gain insight into the adsorption mechanism. The spectra of BP, BPR, and BPRS before and after the adsorption for MB are shown in Figure 5. The peak of 1251 cm$^{-1}$ corresponded to P–P [51]. Generally, the vibration bands of functional groups, interacting with MB, would shift to higher wave numbers after the adsorption was completed, due to furnishing electrons from adsorbate to adsorbent [52]. The peak of 1251 cm$^{-1}$ shifted to higher wave numbers after adsorption; the results meant that the main functional group involved in the adsorption reaction was P–P, which might play critical roles during the adsorption process for MB. In addition, the peaks of S=O in BPRS at 1037 and 1125 cm$^{-1}$ [53] were not observed in the spectra of BP and BPR, which meant the BPR was successfully modified by the sulfonic acid functional group, and the new functional group of S=O was introduced.

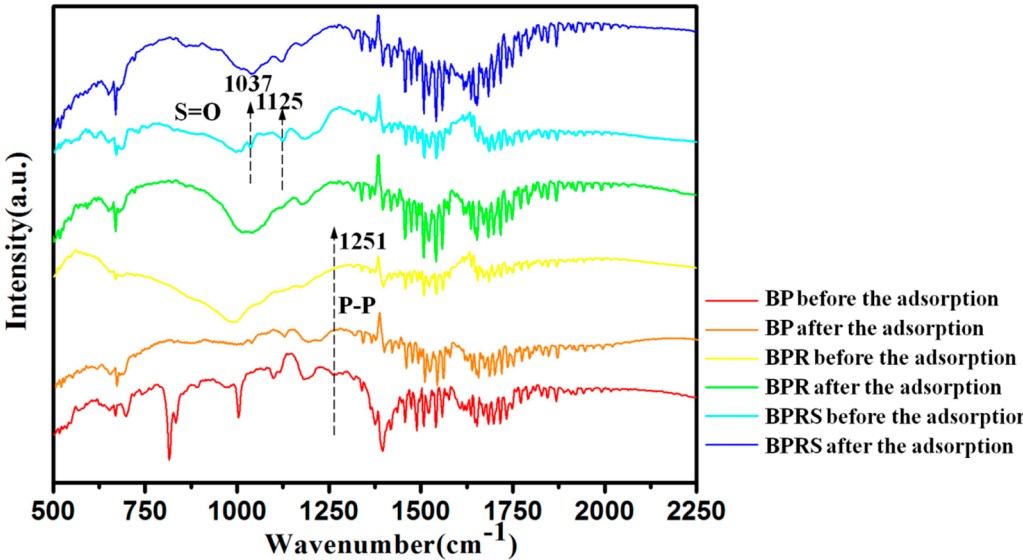

**Figure 5.** FTIR spectra of BP, BPR, and BPRS before and after the adsorption for MB.

### 3.2. Adsorption for MB

### 3.2.1. Adsorption Equilibrium Time

The adsorption equilibrium time of BP, BPR, and BPRS for MB is shown in Figure 6. The adsorption capacities of BPR and BPRS for MB increased rapidly in the first 30 h and achieved equilibrium in approximately 72 h. However, the adsorption equilibrium time of BP for MB was 83 h. The $q_e$ of BP, BPR, and BPRS for MB were 66.98, 86.27, and 99.87 mg·g$^{-1}$, respectively. In previous reports, the $q_e$ of GP nanosheet/magnetite (Fe$_3$O$_4$) composite, UiO-66/nanocellulose aerogels composite, and reduced graphene oxide (rGO) for MB were 43.82, 51.80, and 57.00 mg·g$^{-1}$, respectively [36,54,55]. The results meant that the three BP-based materials were efficient adsorbents for MB compared with other materials. The previous study indicated that the wrinkle of BP could be formed from the interactions between BP and the ionic dyes, which produced new sorption sites between layers, then further induced adsorption [28]. The adsorption capacities of the BP-based materials were increased with the increasing of the wrinkle-induced sorption. However, the adsorption equilibrium time of sulfonated graphene (GS), UiO-66/nanocellulose aerogels composite, and rGO for MB were only 0.5, 6, and 72 h, respectively [36,54,55]. The adsorption equilibrium time of them for MB was shorter than that of the three BP-based materials, because the adsorption induced by wrinkles was a gradual process that took longer. Although wrinkles had a certain promotion effect on adsorption, their adsorption capacity was limited.

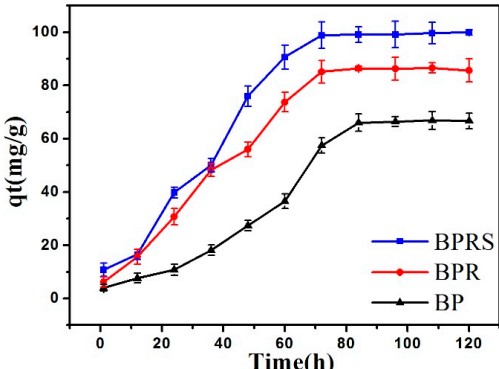

**Figure 6.** Adsorption equilibrium time of BP, BPR, and BPRS for MB.

### 3.2.2. Adsorption Kinetics

The adsorption kinetic curves and parameters of BP, BPR, and BPRS for MB are shown in Figure 7 and Table 2. The points in Figure 7a,c,e were closely distributed on both sides of the line, and the $ln(q_1 - q_t)$ decreased with the increase of time. However, the points in Figure 7b,d,f were sparsely distributed on both sides of the line, and the change of the $t/q_t$ with time had no rules to follow. In addition, th values of $R^2$ were 0.850, 0.914, and 0.933 for the quasi-first-order kinetic model and 0.179, 0.564, and 0.646 for the quasi-second-order kinetic model, respectively, as listed in Table 2. The results indicated that the quasi-first-order kinetics of the adsorption of the three BP-based materials for MB fitted better than the quasi-second-order kinetics. Therefore, the adsorption kinetics of BP, BPR, and BPRS for MB could be explained by the quasi-first-order kinetic model, and the adsorption regulation of the three BP-based materials was consistent. According to the quasi-first-order kinetic model, the adsorption rate could be affected by the solution concentration and the adsorption amount; moreover, the mass transfer resistance within the particles was the limiting factor of adsorption [56]. The results of the adsorption kinetics, fitted better with the quasi-first-order model, confirmed that the adsorption behavior was a diffusion controlled process [55]. Furthermore, both the quasi-first-order model and the quasi-second-order model could fit the kinetic results of rGO for MB well [36], which implied that the adsorption mechanisms of MB onto the three BP-based materials were different from the adsorption mechanism onto rGO. The $q_1$ values (80.891, 95.488, and 120.217 mg·g$^{-1}$) of BP, BPR, and BPRS calculated from the quasi-first-order equation were closer to the obtained value $q_e$ (66.98, 86.27, and 99.87 mg·g$^{-1}$) from the adsorption experiment in this work. The $k_1$ value of BPR (0.070) was larger than that of BP (0.044), as the smaller the block structure of BPR, the shorter distance between MB molecules and adsorbent. However, the time required for adsorption was almost equal, so $k_1$ of BPR was larger. The $k_1$ value of BPRS (0.066) was less than that of BPR (0.070), because the sulfonic acid functional groups on the surface of BPRS would chemically bond with the functional groups of the MB molecules. Compared with the physical adsorption, the slower speed of chemical bonding adsorption was the reason that the $k_1$ value of BPRS was lower than that of BPR.

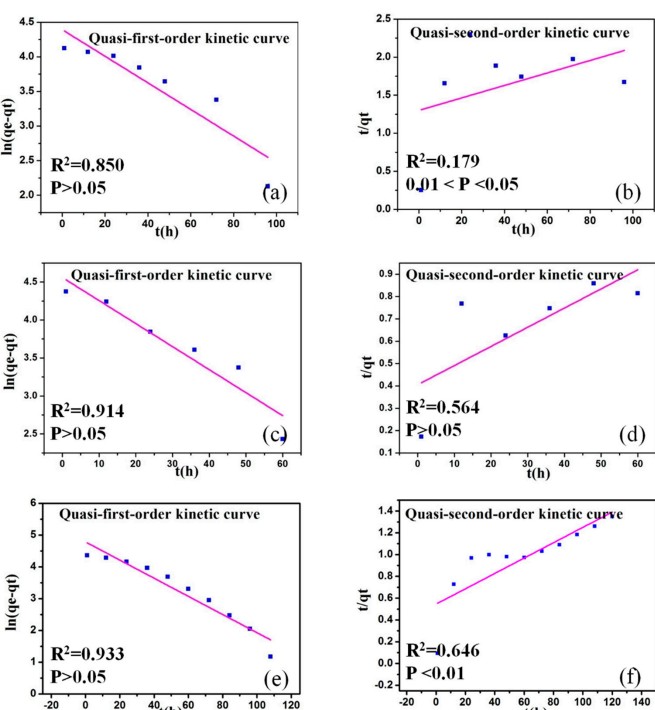

**Figure 7.** Adsorption kinetics for MB: (**a**) quasi-first-order kinetic of BP, (**b**) quasi-second-order kinetic of BP, (**c**) quasi-first-order kinetic of BPR, (**d**) quasi-second-order kinetic of BPR, (**e**) quasi-first-order kinetic of BPRS, and (**f**) quasi-second-order kinetic of BPRS.

**Table 2.** Adsorption kinetic parameters of BP, BPR, and BPRS for MB.

| Adsorbent | Quasi-First-Order Kinetic | | | Quasi-Second-Order Kinetic | | |
|---|---|---|---|---|---|---|
| | $q_1$ (mg·g$^{-1}$) | $k_1$ (h$^{-1}$) | $R^2$ | $q_2$ (mg·g$^{-1}$) | $k_2$ (g·(mg·h)$^{-1}$) | $R^2$ |
| BP | 80.891 | 0.044 | 0.850 | 121.951 | $0.517 \cdot 10^{-4}$ | 0.179 |
| BPR | 95.488 | 0.070 | 0.914 | 116.279 | $0.182 \cdot 10^{-3}$ | 0.564 |
| BPRS | 120.217 | 0.066 | 0.933 | 136.986 | $0.991 \cdot 10^{-4}$ | 0.646 |

### 3.2.3. Adsorption Isotherms and Thermodynamics

The adsorption thermodynamics of BP, BPR, and BPRS for MB at 298.15, 308.15, and 318.15 K are shown in Figure 8a,d,g. Among them, the adsorption efficiency was the best at 318.15 K. Therefore, the adsorption isotherms of the three BP-based materials for MB at 318.15 K were fitted using the Langmuir model (shown in Figure 8b,e,h) and the Freundlich model (shown in Figure 8c,f,i). The corresponding isotherm parameters at 318.15 K are listed in Table 3. The values of $R^2$ were 0.996, 0.999, and 0.981 for the Langmuir model, and 0.895, 0.871, and 0.964 for the Freundlich model, respectively. The results indicated that the adsorption isotherms of the three BP-based materials for MB were fitted better by the Langmuir model. As a result, the adsorption tended toward monolayer rather than multilayer, all adsorption sites were the same, the adsorbed particles were completely independent, the surface of the three BP-based materials was relatively homogeneous, and the adsorption belonged to chemisorption [57,58]. As listed in Table 3, the $q_m$ of MB on BPRS (140.845 mg·g$^{-1}$) was higher than that on BP (84.034 mg·g$^{-1}$) and BPR (91.743 mg·g$^{-1}$) in this work. Although the reported SSA of rGO was 399 m$^2$·g$^{-1}$, the adsorption capacity of rGO for MB was only 136 mg·g$^{-1}$ [36], which was lower than that of BPRS due to its functional group sites. The $k_L$ values of the adsorption for MB by BP, BPR, and BPRS were 7.933, 21.800, and 11.833 L·g$^{-1}$, respectively. The results indicated BPR had the best adsorption affinity for MB, followed by BPRS, and finally BP.

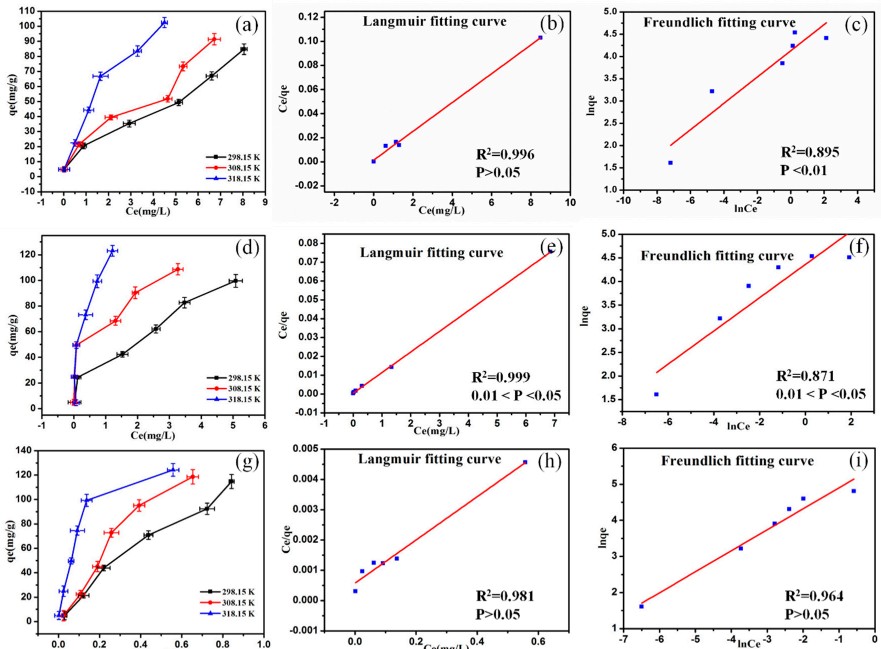

**Figure 8.** (**a**) Adsorption isotherm and thermodynamics of BP, (**b**) Langmuir model of BP at 318.15 K, (**c**) Freundlich model of BP at 318.15 K, (**d**) adsorption isotherm and thermodynamics of BPR, (**e**) Langmuir model of BPR at 318.15 K, (**f**) Freundlich model of BPR at 318.15 K, (**g**) adsorption isotherm and thermodynamics of BPRS, (**h**) Langmuir model of BPRS at 318.15 K, and (**i**) Freundlich model of BPRS at 318.15 K.

**Table 3.** Adsorption isotherm parameters of BP, BPR, and BPRS for MB at 318.15 K.

| Adsorbent | Langmuir Model | | | Freundlich Model | | |
|---|---|---|---|---|---|---|
| | $q_m$ (mg·g$^{-1}$) | $k_L$ (L·g$^{-1}$) | $R^2$ | $K_f$ ((mg·g$^{-1}$)·(mg·L$^{-1}$)$^{-n}$) | $n$ | $R^2$ |
| BP | 84.034 | 7.933 | 0.996 | 62.110 | 3.384 | 0.895 |
| BPR | 91.743 | 21.800 | 0.999 | 78.367 | 2.843 | 0.871 |
| BPRS | 140.845 | 11.833 | 0.981 | 240.760 | 1.719 | 0.964 |

The thermodynamic parameters of MB adsorption by the three BP-based materials are listed in Table 4. $\Delta H^0 > 0$ indicated that the adsorption of the three BP-based materials for MB were endothermic reaction [59]. The total entropy change of the adsorption process was the combined effect of solute adsorption (decreasing entropy) and solvent desorption (increasing entropy), depending on the strength and molecular size of the solute and solvent solid surface [60]. $\Delta S^0 > 0$ indicated that entropy increased in the adsorption process, which implied that the entropy changes caused by the desorption of MB on the surface of the three BP-based materials were greater than those caused by the adsorption. $\Delta G^0$ varied greatly with temperature, because the temperature was a significant physico-chemical parameter, which would change the mobility of the MB molecules, the number of active sites, and the adsorptive forces between the MB molecules and the active sites of the three BP-based materials [61]. $\Delta G^0 < 0$ meant that the adsorption processes of MB from solution to the surfaces of the three BP-based materials were spontaneous. The driving force and affinity during the adsorption processes would strengthen with the $\Delta G^0$ absolute value increasing [62]. The $\Delta G^0$ absolute values of the three BP-based materials were all the highest at 318.15 K, indicating that the best adsorption effects could be obtained at 318.15 K. In addition, the $\Delta G^0$ absolute values for MB by BPRS, BPR, and BP at 318.15 K were 17.424, 12.971, and 8.551 KJ·mol$^{-1}$, respectively. The results revealed that BPRS had the greatest adsorption driving force and affinity for MB, followed by BPR, and finally BP.

**Table 4.** Adsorption thermodynamic parameters of BP, BPR, and BPRS for MB.

| Adsorbent | $\Delta H^0$ (KJ·mol$^{-1}$) | $\Delta S^0$ (KJ·mol$^{-1}$) | $\Delta G^0$ (KJ·mol$^{-1}$) | | |
|---|---|---|---|---|---|
| | | | 298.15 K | 308.15 K | 318.15 K |
| BP | 35.983 | 0.140 | −5.743 | −6.716 | −8.551 |
| BPR | 68.229 | 0.255 | −7.855 | −9.850 | −12.971 |
| BPRS | 68.289 | 0.269 | −12.022 | −14.064 | −17.424 |

### 3.2.4. Desorption

Many studies have shown that there was desorption hysteresis during the desorption process of porous structural materials. Desorption hysteresis refers to a phenomenon where the desorption isotherm and the adsorption isotherm did not coincide under isothermal conditions. The root cause of the desorption hysteresis was the unequal adsorption–desorption rate: the desorption rate was slower and the time required was longer [63]. The desorption hysteresis could be explained by two hypotheses: one was caused by the capillary concentration of molecules in the mesoporous and macroporous materials; the other was caused by the deformation of the pore structure [64,65]. Some researchers believed that it was caused by the rearrangement or agglomeration of the adsorbents [66]. From Figure 9a–c, there were different degrees of desorption hysteresis phenomena on the three BP-based materials. Compared with the adsorption curves, the desorption curves moved to the upper left direction obviously. The results indicated that MB tended to remain in the three BP-based materials, due to the presence of hydrogen bonds between MB and the three BP-based materials, which hindered the diffusion of MB in the adsorbents. It also indicated that there could be the capillary concentration of molecules and the deformation of pore structure in the adsorbents. The as-prepared BP-based adsorbents were not suitable for reuse.

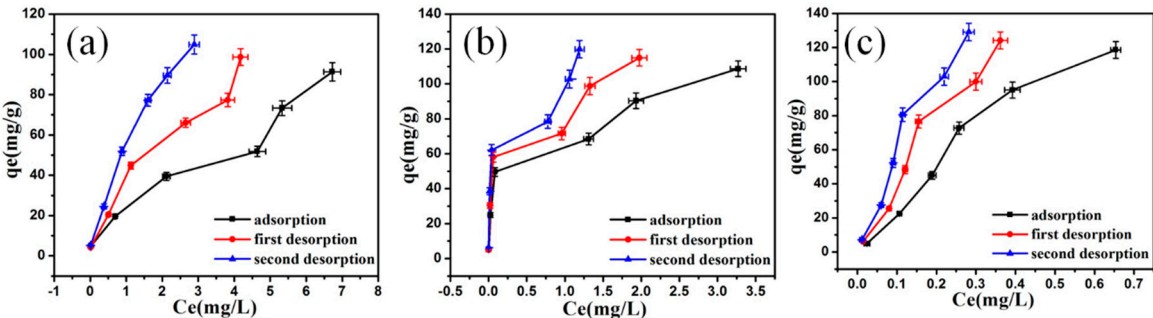

**Figure 9.** Desorption of MB on (**a**) BP, (**b**) BPR, and (**c**) BPRS at 308.15 K.

3.2.5. Influence Factors

Figure 10a–c illustrated the effect of adsorbent dosage on the $q_e$ and adsorption efficiency of the three BP-based materials for MB, respectively. In general, more available and active sites could be generated with the increasing of the adsorbent dosage. With the adsorbent dosage increasing from 1 to 2 mg, the $q_e$ of BP, BPR, and BPRS for MB firstly increased, and then achieved the maximum (66.72, 86.15, and 98.13 mg·g$^{-1}$, respectively). However, as the adsorbent dosage further increased from 2 to 5 mg, the $q_e$ of the three BP-based materials for MB decreased sharply. When the adsorbent dosage increased to a certain amount, there would be some extra adsorption sites, and the adsorption sites would appear "remaining" phenomenon, so that the $q_e$ reduced instead. Simultaneously, with the adsorbent dosage increasing from 1 to 5 mg, the adsorption efficiencies for MB increased from 26.81 to 93.95% by BP, from 27.51 to 95.62% by BPR, and from 44.42 to 99.22% by BPRS, respectively. The results indicated that the optimal adsorbent dosage of BP, BPR, and BPRS for MB (20 mg·L$^{-1}$) was 2 mg in this work. Comparing the three BP-based materials, BPRS had the largest $q_e$ and adsorption efficiency, owing to the dispersibility and adsorption sites of BPRS caused by the sulfonic acid functional groups. BPR had better $q_e$ and adsorption efficiency than BP owing to bigger SSA. Kumar et al. studied the effect of adsorbent dosage on the adsorption of fly ash for MB; the results showed that the adsorption efficiencies ranged from 45.16% to 96.00% with the adsorbent dosage ranging from 8 to 20 g [67]. Zou et al.'s research results also indicated that the adsorption efficiencies for MB ranged from 34.4% to 96.60% with the addition of 1.5 to 5 g modified sawdust [68]. It could be seen that although the adsorption efficiencies of the three BP-based materials for MB were similar to other researchers' results, the required adsorbent dosage was less compared to the previous work, indicating that the BP-based materials could be used as adsorbents to the removal of pollutants in water.

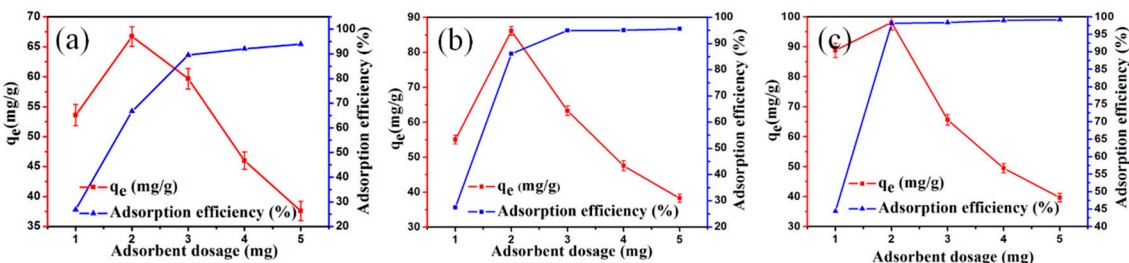

**Figure 10.** Effect of adsorbent dosage on the adsorption for MB by (**a**) BP, (**b**) BPR, and (**c**) BPRS.

Figure 11a–c showed the effect of pH on the $q_e$ and adsorption efficiency of BP, BPR, and BPRS for MB, respectively. pH played an important role during the adsorption process [69]. The changes of pH would cause changes in the chemical form of ionic organic compounds (IOCs), which could change their adsorption characteristics in turn [70]. Due to the two pKa values of MB at 4.52 and 5.85, it could be a cationic (with a positive charge, MB$^+$), anionic (with a negative charge, MB$^-$), and amphiphilic molecule (MB$^0$) [71]. When pH < 4.52, the predominant MB molecule in the solution was MB$^+$ with a positive charge; when 4.52 < pH < 5.85, the neutral MB$^0$ amphiphilic molecule in the solution were

dominant.; when pH > 5.85, the main MB molecule in the solution was $MB^-$ with a negative charge. The previous reports showed that the negative zeta potential of BPR was at $-29.6 \pm 5.03$ mV, which was an ideal value for stabilizing conventional colloidal particles [28,36]. In addition, the dispersions of the three BP-based materials in aqueous had good stability. Therefore, when pH < 4.52, there was a strong electrostatic attraction between MB and BPR, which made the adsorption process easy; when 4.52 < pH < 5.85, the electrostatic attraction between MB and BPR was weak; when pH > 5.85, there was a strong electrostatic repulsion between MB and BPR, making the adsorption process difficult. As BPR still had a high adsorption capacity, there might be additional mechanisms besides electrostatic interaction. The average contact angle of the BPR surface was 57°, indicating that the hydrophobicity of BPR was between graphene oxide (GO, 27°) and GP (90°) [72]. Therefore, the hydrophobic interaction needed to be included. It could also be seen from the experimental results (Figure 11) that the $q_e$ order of BPR for MB was 101.77 mg·g$^{-1}$ (pH = 2.70) > 92.13 mg·g$^{-1}$ (pH = 5.14) > 85.95 mg·g$^{-1}$ (pH = 6.92) > 75.60 mg·g$^{-1}$ (pH = 8.85) > 67.15 mg·g$^{-1}$ (pH = 10.68). The results indicated that the $q_e$ of BPR for MB was affected by electrostatic force. The $q_e$ order of BP for MB was 79.20 mg·g$^{-1}$ (pH = 2.70) > 76.67 mg·g$^{-1}$ (pH = 5.14) > 66.78 mg·g$^{-1}$ (pH = 6.92) > 65.10 mg·g$^{-1}$ (pH = 8.85) > 58.41 mg·g$^{-1}$ (pH = 10.68). The adsorption rule of BP for MB was consistent with BPR, but the $q_e$ was lower than that of BPR due to its smaller SSA, electrostatic force, and hydrophobic force. The $q_e$ order of BPRS for MB was 105.21 mg·g$^{-1}$ (pH = 8.85) > 101.30 mg·g$^{-1}$ (pH = 6.92) > 99.87 mg·g$^{-1}$ (pH = 10.68) > 94.11 mg·g$^{-1}$ (pH = 5.14) > 92.09 mg·g$^{-1}$ (pH = 2.70). The maximum $q_e$ of BP and BPR for MB were at pH = 2.70; nevertheless, the $q_e$ for MB by BPRS was the lowest at the same pH. It could be explained that electrostatic attraction was not the main adsorption mechanism, which was different from the adsorption mechanisms of BP and BPR for MB. The best $q_e$ (105.21 mg·g$^{-1}$) was obtained at pH = 8.85. As BPRS with sulfonic acid functional groups exhibited excellent adsorption effect for MB under alkaline environments, the main adsorption effect was chemical adsorption at this stage. The $q_e$ at pH = 10.68 was lower than that at pH = 8.85; it indicated that hydrophobic interaction was the main adsorption mechanism at this stage. Therefore, from the perspective of the pH changes, the adsorption mechanism of BPRS for MB was gradually changing from electrostatic attraction to chemical adsorption with the pH increasing, and finally to hydrophobic interaction.

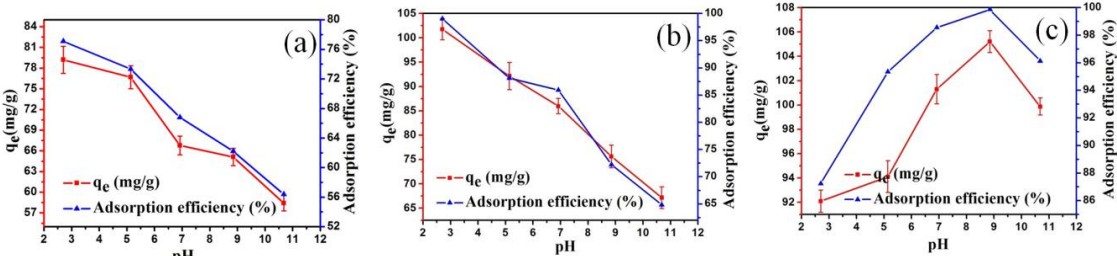

**Figure 11.** Effect of pH on the adsorption for MB by (**a**) BP, (**b**) BPR, and (**c**) BPRS.

With pH increasing from 2.70 to 10.68, the adsorption efficiencies for MB decreased from 77.08% to 56.38% by BP, from 99.04% to 64.82% by BPR, respectively. The adsorption efficiency of BPRS for MB firstly increased, then achieved the maximum (99.87%), and finally decreased. Kumar et al. reported that the adsorption efficiencies of fly ash for MB increased from 36% to 45% with the pH increasing from 2 to 8 [67]. Yagub et al. also studied the effect of solution pH on the adsorption for MB by pine leaves; the results meant that the adsorption efficiencies ranged from 20% to 80% with pH changing from 2 to 11 [73]. In addition, the above reported adsorption efficiencies were lower than those of the three BP-based materials. Hence, the three BP-based materials were effective adsorbents for MB, while BPRS displayed the highest $q_e$. The results suggested that BPRS might be a novel and potential BP-based adsorbent for pollutants in water.

### 3.2.6. Adsorption Mechanism

The $q_m$ of various adsorbents for MB are presented in Table 5. The $q_m$ values of the three BP-based materials in this work were higher than those of most adsorbents, except the three GP-based materials (GO, rGO, and GS) and activated carbon. The BET SSAs of GO, rGO, GS, and activated carbon were 261, 399, 616, and 2032 $m^2 \cdot g^{-1}$, respectively [36,74], which were all higher than those of the three BP-based materials. Research has shown that the deformation of BP was accomplished under a single strain condition, which could control the anisotropic free-carrier mobility and modify the gap size of BP [75,76]. Hence, it could be hypothesized that during the interaction with IOCs, the wrinkle of BP was formed, new adsorption sites between the layers were generated, and the adsorption was further induced. In addition, the electrostatic attraction, hydrophobic interactions, desolvation, and intraparticle diffusion might exist between the three BP-based materials and MB [77–79]. Compared with BP, BPR had the better adsorption efficiency and adsorption kinetic rate, owing to the outstanding adsorption affinity for MB and bigger SSA. In addition, BPRS as a BP-based adsorbent exhibited the most superior adsorption capacity, which was mainly caused by the sulfonic acid functional group sites. Moreover, the dispersibility of BPRS was increasing by modification with sulfonic acid functional groups, which contributed to the adsorption process.

**Table 5.** Maximum adsorption capacities ($q_m$) of MB on various adsorbents.

| Adsorbent | $q_m$ (mg·g$^{-1}$) | SSA (m$^2$·g$^{-1}$) | Ref. |
|---|---|---|---|
| Walnut sawdust | 59.17 | 13.30 | [80] |
| Rice husk | 40.59 | - | [81] |
| Plant leaf powder | 61.22 | - | [82] |
| Sludge ash | 7.99 | 3.70 | [83] |
| Red mud | 2.490 | 21.00 | [84] |
| Sewage sludge | 114.90 | - | [85] |
| Pyrophyllite | 70.42 | 7.03 | [86] |
| Green alga | 40.20 | - | [87] |
| GO | 623.00 | 261.00 | [36] |
| rGO | 136.00 | 399.00 | [36] |
| GS | 906.00 | 616.00 | [36] |
| Activated carbon | 495.00 | 2032.00 | [74] |
| BP | 84.03 | 6.78 | This study |
| BPR | 91.74 | 6.96 | This study |
| BPRS | 140.85 | 7.72 | This study |

## 4. Conclusions

In this work, BP, BPR, and BPRS were prepared, characterized, and their adsorption behaviors for MB were also investigated. The optimal adsorption conditions were as follows: the pH was 2.70 for BP and BPR, and 8.85 for BPRS; the dosage of adsorbents was all 2 mg. The adsorption processes fitted better with the quasi-first-order model. The equilibrium adsorption data of MB could be best modeled using the Langmuir model approach with $q_m$ of 84.03, 91.74, and 140.85 mg·g$^{-1}$ by BP, BPR, and BPRS. Thermodynamic parameters indicated that the adsorption processes were spontaneous, entropy increased, and endothermic reactions. In the processes of desorption, MB tended to be reserved in the three BP-based materials. In addition, the adsorption mechanisms included electrostatic interaction, hydrophobic interaction, intraparticle diffusion, chemical adsorption, sulfonic acid functional group-guided adsorption, and wrinkle-induced adsorption. These research findings could provide references for the synthesis methods of the BP-based materials, and expand their potential applications in adsorption for pollutants in water.

**Author Contributions:** Conceptualization: D.H. and Y.Z.; data curation: J.W.; funding acquisition: J.Q.; methodology: J.W.; project administration: J.Q.; resources: Z.Z.; supervision: J.Q.; validation: H.Y.; visualization:

D.J.; writing—original draft: J.W.; writing—review & editing: J.Q. All authors have read and agreed to the published version of the manuscript.

**Funding:** This research was funded by the National Natural Science Foundation of China [41877364, 41702370, 21707017] and the Jilin Province Science and Technology Development Projects [20200301012RQ].

**Conflicts of Interest:** The authors declare no conflict of interest.

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
