# Peer review of "Comparative Study on the Adsorption Capacities of the Three Black Phosphorus-Based Materials for Methylene Blue in Water"

_sustainability, doi:10.3390/su12208335_

Round 1
Reviewer 1 Report
The work has an optimal degree of innovation. The work is adequate in the current form and it is. Only a small revision is needed, in particular in paragraph 2.3, I suggest the authors to insert the chemical-physical characteristics of the water sample prepared with MB. This information is important for subsequent work, and for comparative analyzes of other studies.
Reviewer 2 Report
Presented paper deals with quite important issue - new adsorbent material. The studies were properly planned and described. Authors presented results in clear way and compared them with the results obtained by other autors. The disscussion of the results seems to be correct and suffiecient.
Reviewer 3 Report
This manuscript reports three black phosphorus-based adsorbents for methylene blue in aqueous solution. The adsorption isotherm and kinetics were systematically studied and analyzed. Although the field of dye adsorption materials is already a crowded one, this manuscript presents interesting results and may have fair contributions to the field.
However, the reviewer has one major concern: how can the as-prepared BP-based adsorbents be reused/regenerated? Have the authors consider the desorption of MB? This is of practical importance for the toxic compound adsorption in water/wastewater treatment.
Additional comments are shown in the following:
- Please do not use the abbreviation (BP) in the title.
- line 140, Although pseudo-first and -second-order rate equations have been widely used for adsorption kinetic data fittings, the misapplications of those equations have been found (AIChE Journal, 64(5) 2018, 1793, DOI 10.1002/aic.16051). I suggest the authors add a short discussion about this issue.
- line 153 to 156, for Kc in equation (6), it should be a unitless parameter. In fact, the thermodynamic equilibrium constant without unit is not determined by equation (7), either. Please check this ref (Yu Liu, J. Chem. Eng. Data 2009, 54, 1981–1985) for more details, and correct the errors accordingly.
- line 167, the reviewer can not get such conclusion that "block structure of BP was thicker than BPR and BPRS" from the SEM pictures.
- Line 177, please add refs.
- Line 200, how can the authors determine that there is a hierarchically porous structure in these materials?
- In Figure 6 and other adsorption-related data, why the error bar is not shown? Were the adsorption experiments only done once? Is the data reproducible?
- Tables 3 and 4, there is a format issue. ΔS(kJ·mol-1) should be in one line.
Reviewer 4 Report
The study deals with the capacity of three nanomaterials based to adsorb polluting compounds in dye wastewater.
The topic is interesting, but the authors failed in properly justifying the novelty of the paper and its contribution to the scientific knowledge in this field. Moreover, the quality of paper, although being quite satisfying, has to be improved, following the suggestions I gave in the commented MS in attachment.
Finallym English requires a substantial revision.

Round 2
Reviewer 4 Report
The authors have replied to the reviewers' comments and now the paper is improved compared to the previous version. I have no more suggestions or corrections.